# Isolation and Characterization of the Genes Involved in the Berberine Synthesis Pathway in Asian Blue Cohosh, *Caulophyllum robustum*

**DOI:** 10.3390/plants12071483

**Published:** 2023-03-28

**Authors:** Nam-Il Park, Neha Samir Roy, Yeri Park, Beom-Soon Choi, Mi Jin Jeon, Ji Yeon Oh, Bo-Yun Kim, Young-Dong Kim, Yong-In Kim, Taeyoung Um, Hwan Jong Kwak, Nam-Soo Kim, Soonok Kim, Ik-Young Choi

**Affiliations:** 1Department of Plant Science, Gangneung-Wonju National University, Gangneung 25457, Republic of Korea; 2Agriculture and Life Sciences Research Institute, Kangwon National University, Chuncheon 24341, Republic of Korea; 3Next Bio Information Technology, Bodeumkwan 504, Kangwon National University, Gangwondaehakgil-1, Chuncheon 24341, Republic of Korea; 4Microorganism Resources Division, National Institute of Biological Resources, Incheon 22689, Republic of Korea; 5Plant Resources Division, National Institute of Biological Resources, Incheon 22689, Republic of Korea; 6Department of Life Science, Multidisciplinary Genome Institute, Hallym University, Chuncheon 24252, Republic of Korea; 7On Biological Resource Research Institute, Chuncheon 24239, Republic of Korea

**Keywords:** *Caulophyllum robustum*, blue cohosh, BIA, berberine, transcriptomes

## Abstract

*Caulophyllum robustum*, commonly named Asian blue cohosh, is a perennial herb in the family Berberidaceae. It has traditionally been used for folk medicine in China. We isolated berberine from the leaves, stem, roots, and fruits of *C. robustum*, and this is the first report on berberine in this species. Transcriptome analysis was conducted for the characterization of berberine biosynthesis genes in *C. robustum*, in which, all the genes for berberine biosynthesis were identified. From 40,094 transcripts, using gene ontology (GO) analysis, 26,750 transcripts were assigned their functions in the categories of biological process, molecular function, and cellular component. In the analysis of genes expressed in different tissues, the numbers of genes in the categories of intrinsic component of membrane and transferase activity were up-regulated in leaves versus stem. The berberine synthesis genes in *C. robustum* were characterized by phylogenetic analysis with corresponding genes from other berberine-producing species. The co-existence of genes from different plant families in the deepest branch subclade implies that the differentiation of berberine synthesis genes occurred early in the evolution of berberine-producing plants. Furthermore, the copy number increment of the berberine synthesis genes was detected at the species level.

## 1. Introduction

*Caulophyllum* is a small genus of perennial herbs in the family Berberidaceae. The genus contains only three species that are grown in the wild in eastern Asia and North America. Of the three species identified, *C. thalictroides* and *C. giganteum* are natives in eastern North America, and *C. robustum* is native to Northeast Asia, including China, Korea, and Japan [1]. The common name for the *Caulophyllum* species is blue cohosh, from the North American natives who used it for the treatment of amenorrhea, dysmenorrhea, and rheumatic symptoms [2]. The extracts are still used by midwives to aid labor in the United States. Root powders or liquid extracts from blue cohosh are sold as dietary supplements and labeled as a woman’s tonic or simply an herbal supplement [3]. However, safety concerns have arisen regarding the effect of specific constituents of blue cohosh on humans and animals, especially during pregnancy [4].

*C. robustum* is an endemic perennial plant species in northeast Asia, including north China, Korea, Japan, and east Siberia in Russia, including Sakhalin [5]. It grows in the wild, flowers in June—July, and bears blue and round fruit 6–7 mm in diameter (http://www.nature.co.kr, accessed on 12 December 2022). It is known in Chinese as *Hong Mao Qi*, and extracts from the roots and rhizomes have been widely used as folk medicine in China for the treatment of menstrual irregularity and stomach ache [6]. Various phytochemicals have been found in the extracts of *Caulophyllum* plants, and the bioactivities of the *Caulophyllum* species include anti-inflammatory [7,8], antibacterial [9], and anti-rheumatism activities [10]. It has also been used to ease childbirth [11]. The known phytochemicals in *Caulophyllum* are alkaloids, triterpene saponins, fatty acids, and sterols (reviewed in Rader and Pawar 2013; Xia et al. 2014, [4,6] references therein).

Benzylisoquinoline alkaloids (BIAs) are very diverse, containing approximately 2500 molecules, including morphine, codeine, sanguinarine, and papaverine [12,13]. However, BIAs have been identified in only a few plant families, including Berberidiceae, Papaveraceae and Ranunculaceae in Ranuculales, Fabaceae in Fabales, Magonoliaceae in Magnoliales [13,14], and Nelumbonaceae in Proteales [15]. Berberine is one of the organic molecules found in heteropentacyclic compounds in the protoberberine group (http://pubchem.ncbi.nlm.gov/compound/2353, accessed on 12 December 2022). The pharmacological activities of berberine are highly diverse, such as in the treatments of type II diabetes, various types of cancer, vascular diseases, oral diseases, various metabolic syndromes, and more (reviewed in Imenshahidi and Hosseinzadeh 2018 [16], references therein).

Many alkaloids have been reported as being present in the genus *Caulophyllum*. Li et al. found 11 alkaloid molecules in *C. robustum* using the binary chromatographic fingerprints of a high-performance liquid chromatography/diode array detector and gas chromatography/mass spectrometry analysis [17]. In a review of the phytochemicals found in the genus *Caulophyllum*, Xia et al. listed 22 alkaloid molecules, including *N*-methylcytosine, lupinine, taspine, and several types of caulophyllines, as being present in root and rhizome extracts [6]. However, no report is available regarding the presence of berberine in the genus *Caulophyllum*. We found berberine in the leaves, stem, roots and fruits of *C. robustum*. We also carried out transcriptome analysis on *C. robustum*. The current report contains the characterization of berberine synthesis genes in *C. robustum* along with a comparative analysis of those genes with genes in other berberine-producing plants.

## 2. Materials and Methods

### 2.1. Plant Materials

A whole plant of *C. robustum* at flowering stage was collected from Yanggu-gun, Gangwon Province, South Korea (38.2577463 N, 128.1003539 E). Table 1 lists other plant species that were used to mine the genes in the berberine biosynthesis pathway. Transcriptomes of other species were downloaded from NCBI.

### 2.2. Berberine Extraction and Quantification

Tissues were collected three times during 15–19 June 2021. The freeze-dried tissues were ground with a mortar and pestle, and berberine was extracted from the fine powder. The tissues were freeze-dried with 70% ethanol (1 g/10 mL) in the dark for 24 h at room temperature. After dilution to 10,000 mg/L, the berberine content was measured using a Shimadzu Prominence HPLC system (Shimadzu, Kyoto, Japan) equipped with a diode array UV–vis detector for monitoring at 280 nm. A C18 column (250 × 4.6 mm, 5 μm, Waters, Ireland) was used for separation by binary gradient elution using solvent A (water containing 0.1% formic acid) and solvent B (acetonitrile containing 0.1% formic acid). The flow rate was set at 1.0 mL/min as follows: 0 min, 0% B; 5 min, 0% B; 10 min, 30% B; 25 min, 40% B; 30 min, 50% B; 35 min, 0% B; and 40 min, 0% B. The injection volume was 10 μL and the column temperature was 40 °C.

One-way Analysis of Variance (ANOVA) and Duncan’s multiple range tests were used to identify significant differences (*p* < 0.05), using Statistical Product and Service Solutions (SPSS ver. 26, Chicago, IL, USA).

### 2.3. RNA Extraction and Transcriptome Sequencing

Transcriptome sequencing was conducted on the fruit, leaves, and stem of *C. robustum.*

Total cellular RNAs were extracted from fresh tissues with the Hybrid-RTM kit (Genes All Biotechnology Co., Seoul, Republic of Korea). A cDNA library was constructed using the TruSeq RNA Sample Prep Kit v2 (Illumina). Paired-end sequencing was carried out with the Illumina HiSeq 4000 at Macrogen, Inc. (Seoul, Republic of Korea). Low-quality reads (Phred score < 20), short reads (<50 bp), empty nucleotides (N at the end of reads), and adapter sequences were removed using Trimmomatic software [18]. Quality control before and after trimming was performed using FastQC software (https://www.bioinformatics.babra.ham.ac.uk/projects/fastqc/, accessed on 12 December 2022).

### 2.4. De Novo Assembly

Transcriptome sequences were assembled with the *de novo* RNA-Seq Assembly Pipeline using Trinity v2.13.2 with the default option (https://github.com/trinityrnaseq/trinityrnaseq/wiki, accessed on 12 December 2022). Duplicated contigs were removed using CD-HIT-EST software (http://weizhong-lab.ucsd.edu/cd-hit/, accessed on 12 December 2022). For low coverage removal, the raw reads in the contig assembly were remapped and the contigs with less than 10 hits were removed using Samtools v1.13 (http://www.htslib.org/, accessed on 12 December 2022) with the default option. The coding region was, then, finally confirmed using TransDecoder (v3.0.1) (https://github.com/TransDecoder/TransDecoder/wiki, accessed on 12 December 2022) [19]. Short reads were mapped to the assembly using the Burrows–Wheeler Aligner (BWA) (https://github.com/lh3/bwa, accessed on 12 December 2022) [20]. The fragments per kilobase per million (FPKM) was calculated to select fragments higher than 1 FPKM in value to be used as a reference sequence.

### 2.5. Functional Annotations

The transcriptome sequences were blasted on the known public databases InterProScan in the European Bioinformatics Institute (EBI) and NR in NCBI using the Basic Local Alignment Search Tool (BLAST) program with the cut-off parameters e-value 1e^−4^ and >70% similarity. The candidate transcripts with InterProScan and BLASTP hits were sorted to import to the Blast2GO suite 6.0 (https://www.blast2go.com/, accessed on 12 December 2022, BioBam Bioinformatics SL, Valencia, Spain). Gene ontology (GO) analyses were then performed.

### 2.6. Isolation of Berberine Synthesis Genes

Berberine synthesis genes were reported in *Berberis koreana* [21]. We used the protein sequences of the berberine synthesis genes in *B. koreana* for query in the transcriptome sequences in *C. robustum* using the NBLAST program [22]. The genes isolated were blasted again in NCBI database to be sure their functions. We also isolated the corresponding sequences from the NCBI genome databases of those plant species listed in Table 1.

### 2.7. Phylogenetic Analysis

Phylogenetic analysis was performed using the protein sequences. The protein sequences were aligned using ClustalW software and the phylogenetic tree was constructed with the neighbor-joining method using MEGA v5.0 with 1000 bootstraps. A maximum likelihood phylogenetic tree was built using MEGA X (v10.2.4) [23].

## 3. Results

### 3.1. Berberine Contents

We quantified the berberine content in the leaves, stem, roots, and fruit of *C. robustum* (Figure 1). The berberine content in leaf tissue was more than 10 times higher than in the other three tissues, that is, 58.51 ± 3.39 ug/g in the leaf, 5.83 ± 0.24 ug/g in the stem, 2.60 ± 1.33 ug/g in the root, and 3.23 ± 0.09 ug/g in the fruit per 1 g of tissue.

### 3.2. Transcriptome Sequencing

We obtained approximately 1.97 × 10^8^ reads from the three transcriptome libraries (Appendix A). The percentage of clean reads from the raw reads was approximately 91.7% in the three libraries. Briefly, the number of clean reads was approximately 5.3 × 10^7^ in the fruit, 6.4 × 10^7^ in the leaves, and 6.7 × 10^7^ in the stem of *C. robustum*. From the cleaned read data, the number of mapped reads were 3.7 × 10^7^, 4.6 × 10^7^, and 4.3 × 10^7^ in the fruit, leaves, and stem, respectively (Appendix A).

The de novo assembly was carried out by pooling the three library results (fruit, leaf, stem). (Appendix A). The number of contigs obtained was 252,149, which covers 1.97 × 10^8^ bp. After cleaning by removing redundancy, the number of contigs was 41,094, which covers 5.4 × 10^7^ bp. The average length of the contigs was 783 bp before cleaning and 1338 bp after redundancy cleaning.

### 3.3. Functional Annotation and Gene Ontology (GO)

The functions of the obtained transcripts were analyzed by mapping against the InterProScan and NR databases. Of the 41,094 transcripts, 24,840 (60.4%) and 30,210 (73.5%) were mapped against the InterProScan and NR databases, respectively (Figure 2A). In the NR database, the highest matching was with *Macleaya cordata* (3884), *Aquiegia coerulea* (3113), and *Tetracentron sinense* (2277), followed by *Coptis chinensis, Thalictrum thalictroides*, and *Nelumbo nucifera* (lotus) (Figure 2B).

In the GO analysis, 26,750 transcripts were assigned into the three major classification categories: biological process, molecular function, and cellular component, in which, 16,224, 18,160, and 15,282 transcripts were of biological process, molecular function, and cellular component, respectively (Figure 3).

### 3.4. Differences of the Genes Expressed among Leaves, Stem, and Fruit in C. robustum

Of the 41,094 transcripts obtained in *C. robustum*, the numbers of transcripts expressed in the leaves, fruit, and stem were 37,108, 35,597, and 36,608, respectively (Appendix A). The expression was analyzed using a vis-à-vis comparison between the tissues (Table 2). In a comparison between the fruit and leaves, 761 genes were up-regulated and 1135 genes were down-regulated in the fruit compared with the leaves. In leaves versus stem, the number of up-regulated genes in the leaves was higher than in the stem. GO analysis of the differentially expressed genes in the tissues revealed that the genes related to the membrane and the intrinsic component of the membrane categories were the highest in number in both up- and down-regulation (Figure 4). In leaves versus stem, 326 genes and 105 genes were up- and down-regulated, respectively, in the leaves, in which, a high number of genes in the categories of intrinsic component of membrane and transferase activity were up-regulated, and the genes related to response to stress and plasma membrane were down-regulated (Figure 4). In fruit versus stem, 536 and 768 genes were up- and down-regulated in the leaves, respectively (Table 2). Of the up-regulated genes, the intrinsic component of membrane genes showed the highest peak. Of the down-regulated genes, the number of annotated genes was extensive, but the down-regulated genes were mostly related to metabolite synthesis, including the BIA metabolic process and the BIA biosynthetic process (Figure 4).

### 3.5. Isolation of Berberine Synthesis Genes

Figure 5 shows the berberine synthesis pathway and Table 3 lists the number of berberine synthesis genes in six berberine-producing medicinal plants mined from the NCBI database. Protein sequences of the berberine synthesis enzymes of the six plants are shown in Appendix A. We found the protein sequences of most of the genes in the six species, except NCS and STOX in *P. nudicaule*. The number of copies of the berberine synthesis genes ranged from one to twelve, depending on the genes and species. Protein sequences of the berberine synthesis genes of the Table 3 are in the Appendix A.

NCS is an important enzyme in BIA synthesis because it catalyzes the condensation reaction of dopamine and 4-hydroxyphenyl acetaldehyde to yield (*S*)-norcoclaurine, which is the first common precursor in the synthesis of myriad BIA molecules. The copy numbers of the NCS were ten, eleven, five, six, and one in *C. robustum*, *B. koreana*, *N. nucifera*, *C. chinesis*, and *M. cordata*, respectively. We could not find the NCS matching sequences in the transcriptome database for *P. nudicaule*. Phylogenetic analysis was carried out with these NCSs, while the NCS from *M. cordata* was excluded because it was a partial sequence with only 81 amino acids (Figure 6A). The 32 NCSs from four species were separated into two groups: I and II. The *B. koreana* NCS_2 was out-grouped from the others. Of the ten NCSs from *C. robustum*, nine were placed in group I and one NCS was in group II. Group I contained NCSs from Ranunculaceae and Berberidaceae, in which the NCSs from a species formed mostly separated subclades. Group II contained NCSs from Berberidaceae and Nelumbonaceae. Unlike the NCSs in group I, the NCSs from different species were mixed in subclades.

Sources of the transcriptomic information: *C. robustum*: current research, *B. koreana*: Roy et al., 2022 doi:10.3390/plants10071314 [21], *N. nucifera*: https://www.ncbi.nlm.nih.gov/protein/?term=txid4432[Organism:noexp], accessed on 12 December 2022, *C. chinensis*: https://www.ncbi.nlm.nih.gov/protein/?term=txid261450[Organism:exp], accessed on 12 December 2022, *P. nudicaule*: https://www.ncbi.nlm.nih.gov/ipg/?term=txid74823[Organism:noexp], accessed on 12 December 2022, *M. cordata:* https://www.ncbi.nlm.nih.gov/protein/?term=txid56857[Organism:noexp], accessed on 12 December 2022.

Berberine bridge enzyme (BBE) is the key rate-limiting enzyme in the berberine synthesis pathway [2,24]. We found four copies in the transcriptomes of *C. robustum* and two to five in other berberine-producing plants (Table 3). There were five copies in the transcriptomes of *B. koreana*, which belongs to the same Berberidaceae plant family as *Caulophyllum*. We analyzed the phylogenetic relationships among the BBEs listed in Table 3 as well as a few more BBEs from other species that were mined from the NCBI protein database (Figure 6B). Short protein sequences were excluded from the analysis so that the phylogenetic analysis included 32 BBEs of 11 species from three families. The analyzed BBEs were divided into two groups, I and II, which were further divided into four subgroups. In each subgroup, the BBEs from different plant families were co-existed, except for subgroup I-III, which contained BBEs from *B. koreana* out-grouped to the two sister subgroups I-I and I-II.

### 3.6. Phylogenetic Analysis of Methyltransferases and CYP450 Monooxygenases

There were four methyltransferases (6OMT, CNMT, 4OMT, and SOMT) and two CYPs (CYP80B3 and CYP719A21) in the berberine synthesis pathway. Phylogenetic analysis was carried out with the four methyltransferases and two CYP subfamilies.

The analyzed 58 methyltransferases from 10 plant species were divided into five groups (Figure 7). The CNMTs were placed into a group V that was an out-clade to the *O*-methyltransferases. Three 4OMTs of *N. nucifera* formed a group IV that was an out-clade to groups I (6OMTs and 4OMTs), II (6OMTs), and III (OMT of *A. thaliana* and 4OMTs); while CNMTs and SOMTs were clustered in their own subclade, the 4OMTs and 6OMTs split into two groups. The 4OMTs split into two groups, in which the three 4OMTs from *N. nucifera* formed a single cluster that was out-grouped to the other *O*-methyltransferases. *A. thaliana* OMT clustered with the SOMT clade (group IV) that was tied with two sister clades, in which one clade consisted of 6OMTs and 4OMTs (group I) and another clade comprised only 6OMTs (group II). In each clade of the deep branch, the CNMT and *O*-methyltransferases consisted of enzymes from different species from different families. In 6OMT in group I, for instance, the 6OMTs were from *C. robustum* (Berberidaceae, Ranunculales), *B. koreana* (Berberidaceae, Ranunculales), *C. chinensis* (Ranunculaceae, Ranunculales), *M. cordata* (Papaveraceae, Ranunculales), *P. papaver* (Papaveraceae, Ranunculales), and *N. nucifera* (Nelumbonaceae, Proteales). Because the 6OMTs split into two groups, multiple sequence analysis was carried out with 6OMTs from *C. robustum* and *B. koreana* in Berberidaceae (Figure 7). There were six and seven copies of 6OMTs in *C. robustum* and *B. koreana*, respectively. Polypeptide sequence conservation of group I 6OMTs was higher than that of group II 6OMTs, including the S-adenosyl-L-methionine binding (conserved region I) and metal binding (conserved region IV) sites (Appendix A).

There were 184 CYP genes that fell into 31 families of CYP monooxydases in the transcriptomes of *C. robustum* (Table 4). The family was identified by the NCBI nomenclature. The CYP71 and CYP72 had predominantly high number of genes with 27 and 22, respectively. CYP78 amd CYP94 had 10 and 14 genes, respectively. In each gene, CYP71A1 and CYP72A had 13 copies and 8 copies, respectively, and the rest genes had variable number of copies from one to six. Of the 184 CYP genes of *C. robustum*, 21 genes were found in the transcriptome of *B. koreana* [21].

Of the various CYP enzymes, CYP80 and CYP719 are involved in the berberine synthesis and we found single gene in both CYP genes in the transcriptomes of *C. robustum*. Thus, we carried out phylogenetic analysis with the CYP80 and CYP719 enzymes from berberine producing plants except of the *Combretum micranthum*. The phylogenetic analysis revealed that CYP80 and CYP719 were distinguished by forming separate clades (Figure 8). As expected, the CYP80 and CYP719 of *C. robustum* showed closest relationship by forming a deepest branch, subclade I-II in group I and IV-III in group IV, respectively.

## 4. Discussion

*C. robustum* (Asian blue cohosh) has long been traditionally used in folk medicine in China [6,7]. It is still used in the United States by midwives to aid labor [3]. Xia et al. listed 22 phytochemicals identified in root and rhizome extracts from the *Caulophyllum* species; however, berberine was not included in the list [6]. Berberine is a major bioactive compound among diverse benzylisoquinoline alkaloids (BIAs) in the species in Berberidaceae [25]. *Caulophyllum* is a small genus in the Berberidaceae family [1], and this is the first report on berberine in the *Caulophyllum* species. We found that the leaves contained considerably more berberine than other tissues. However, the plant parts used in folk medicine are mostly roots, rhizomes, and stems [4,7,8]. Thus, *C. robustum* leaf may be a good source for isolating berberine for the pharmaceutical industry.

The transcriptome mapping frequency was 64.4% and 73.5% of the InterProScan and NR databases in our results. These values are higher than the 69.9% NR matching in the transcriptomes of *Berberis koreana* [21], 46.3% in *Coptis teeta*, and 42.64% in *Coptis chinensis* [26]. In the NR database, the top six species showing over 1000 matches are known to produce BIA alkaloids [21,26,27,28], except *Tetracentron sinense*. BIA alkaloids are limited in some plant families, such as Berberidaceae, Papaveraceae, and Ranunculaceae in Ranunculales [14] and Nelumbonaceae [15]. The five species showing the highest matching belong to these limited plant families. There have been no reports on the BIAs in *T. sinense*, which is a basal eudicot species lacking a vessel element [29] in the family Trochodendraceae in Trochodendrales. Roy et al. reported the highest NR matching between the transcriptomes of *B. koreana* and *Nelumbo nucifera* (lotus) [21]; whereas in our results, the matching between the transcriptomes of *C. robustum* and *N. nucifera* was lower than other species in Berberidaceae, Papaveraceae, and Ranunculaceae. Significant sequence homologies were reported between the transcriptomes of *Coptis deltoidea* and *N. nucifera* [30], and 1653 transcripts were still matched between *N. nucifera* and *C. robustum* in our results. Many bioactive components, including BIAs, have been found in *N. nucifera*, and all parts of the lotus plant have been used in Chinese traditional medicine [31,32].

In our analysis, the GO classification of 65.1% was higher than the 53.1% in *B. koreana* [21], but lower than those of the *Coptis* species, with 77.8% in *C. deltoidea* [30], and 70.5% in *C. chinensis* [26]. In the biological process category, most of the functions were related to metabolic processes. In the transcripts classified as biological processes, most of them were related to metabolic processes in which two groups with over 8000 transcripts were involved in organic substance metabolic processes and nitrogen compound metabolic processes. Alkaloids are natural organic cyclic compounds that contain at least one nitrogen atom [33]. In the molecular function category, the top two groups were organic cyclic compound binding and heterocyclic compound binding, which is reasonable as all alkaloids are heterocyclic and organic cyclic compounds. The reason that oxidoreductase activity and transferase activity groups were also found in high numbers could be related to the fact that the biosynthetic pathway for the BIA molecules was mediated by several methyltransferases and CYPs [21,34]. The GO analysis of the genes expressed among different tissues supports the finding of a higher berberine content in *C. robustum* leaf than in the stem, fruit, or root. In the comparison between the fruit and leaves, the up-regulated genes in the leaves included transferase and oxidoreductase activities. In the comparison between the leaves and stem, the up-regulated genes in the leaves were directly related to functioning in the BIA metabolic process, BIA biosynthetic process, oxidoreductase activity, and monooxydase activity, although the numbers were less than 10 in each subcategory (Figure 4).

BIA biosynthesis starts with the condensation of dopamine and 4-hydroxyphenylacetaldehyde, both derived from *L*-tyrosine, to become (*S*)-norcoclaurine, which then undergoes enzyme-mediated stepwise reactions including oxidation, reduction, methylation, acetylation, and decarboxylation, resulting in structural changes in the molecules [35]. The biosynthetic steps and enzymes have been well elucidated in the *Coptis* species [26], *Berberis* species [21,36], and other plants [34]. We identified all biosynthetic enzymes for berberine synthesis in the transcriptomes of *C. robustum*. The number of copies of genes encoding each enzyme was variable, with the highest copy of 10 in NSC. We also isolated berberine synthesis genes from the transcriptomes in the NCBI database in other berberine-producing plants, which revealed that the copy numbers of each enzyme-encoding gene were highly variable in each species. The copy numbers of the berberine synthesis genes were also variable among the three closely related species in the genus *Berberis* [36]. In contrast to our results, half of the genes identified in the BIA synthesis were identical copies between *Coptis teeta* and *C. chinensis* [26].

NCS is a key enzyme for the biosynthesis of all BIA alkaloids as it catalyzes the condensation of dopamine and 4-hydroxyphenylacetaldehyde (4-HPAA) to form (*S*)-norcoclaurine, which is the entry molecule for BIA synthesis [34,36,37]. *C. robustum* has ten copies of NCS, which is the highest number among the BIA synthesis enzymes in *C. robustum*. In phylogenetic analysis, the group II NCSs might be ancestral to the group I NCSs because group II consists of families from Berberidaceae and Nelumbonaceae. The Nelumbonaceae belongs to the order of Proteales, whereas Ranunculaceae and Berberidaceae are in the order of Ranunculales. In addition, the NCSs in the deep branch clade were strictly with a single species in group I, but not in group II. NCSs from *Thalictrum flavum*, *Papaver somniferum*, and *Coptis japonica* shared significant sequence similarity, suggesting that NCS might be a member of the pathogenesis-related 10/Bet V1 protein family [37]. Thus, functional differentiation may be ensued by limited sequence modification. In a phylogenetic analysis of those NCSs with the known pathogenesis-related 10/Bet V1 protein, the (*S*)-norcoclaurine synthesis enzymes showed clustering together, so the authors suggested a monophyletic origin for NCSs. Our results showing the co-existence of NCSs from different plant families also support the monophyletic origin theory.

The berberine bridge enzyme (BBE) has a synonym (*S*)-reticuline oxidase as it catalyzes the conversion of (*S*)-reticuline to (*S*)-scoulerine by an oxidative ring closure conversion of (*S*)-reticuline by C-C bond, referred to as the berberine bridge [38]. As (*S*)-scoulerine is a common intermediate molecule for the biosynthesis of sanguinarine, berberine, and noscapine [34], the conversion of (*S*)-reticuline to (*S*)-scoulerine is the rate-limiting step for BIA production in many BIA-producing plants [28]. There were four copies of BBE in the *C. robustum* transcriptomes. In the phylogenetic study, thirty-two BBE enzymes from eleven species in three families were split into two groups, which were further divided into four subgroups, in which the BBEs from different plant families co-existed. BBEs from a single genus grouped together at the deepest subclade in most cases, but odd clustering at the deepest cluster was observed, such as *C. robustum* G48747T and *B. stolonifera* AAD17487 in subgroup II-I, and *B. lycium* AWH63007.1 and *P. somniferum* in subgroup II-II with robust bootstraps. BBE-like enzymes are a multigene family of FAD-linked oxidase (pfam08031) [39] present in plants, fungi, and bacteria [38]. In plants, the BBE-like proteins were identified in wide range of plant families as mediating the production of alkaloids [38]. BBEs from different plant families co-existed in the subgroups in our phylogenetic analysis. Thus, the BBEs in the current study represent the monophyletic enzymes in the BIA synthesis pathway among the diverse BBE-like proteins.

A diverse group of methyltransferases mediate the production of naturally produced small molecules by adding methyl groups to *S*, *N*, *O*, or *C* atoms [40]. Various *O*- and *N*-methyltransferases were identified in the production of alkaloids in plants [41]. For berberine synthesis, one *N*-methyltransferase (CNMT) and three *O*-methyltransferases (4OMT, 6OMT, SOMT) are known to be involved in berberine synthesis in the *Berberis* species [21]. In the current study, we identified one copy of CNMT, four copies of 4OMT, six copies of 6OMT, and three copies of SOMT in the transcriptomes of *C. robustum*. In the phylogenetic analysis, the cluster of each methyltransferase and CNMT and the co-existence of methyltransferases from different species in the deep branch subclade indicated that each methyltransferase was monophyletic, evolving independently from each other after diverging from ancestral methyltransferase. Liu et al. demonstrated the whole gene duplication and diversification of protoberberine synthesis genes in *C. chinensis* from the whole sequence data [40], which can account for the divergence of the 6OMT group I and 6OMT group II in our result. The two subgroups of 6OMTs may have derived from the duplication of the 6OMT encoding gene. The duplication might have occurred in a very early stage of plant evolution as a result of the coexistence of *N. nucifera* (Proteales, Nelumbonaceae) and *C. chinensis* (Ranunculales, Ranunculaceae), along with *C. robustum* and *B. koreana* (Ranunculales, Berberidaceae) in the 6OMT group II clade. The 6OMT group I also constituted enzymes from different plant families from different orders. Ibrahim et al. identified five conserved sequence motifs in *O*-methyltransferases [42]. We identified these conserved motifs in the enzymes of both 6OMT groups, but each group showed slightly different sequences in each motif, alluding to the fact that they may have undergone some sequence diversification, but conserved the sequences’ functional motifs, such as motif I for *S*-adenosyl *L*-methionine and motif IV for metal binding.

Cytochrome P450 (CYPs) are a monooxygenase superfamily containing heme as a cofactor [43]. CYP genes are estimated to comprise approximately 1% of the plant genome [44,45] and plant CYPs are well archived in the Plant CYP website (http://drnelson.uthsc.edu/biblioD.html, accessed on 12 December 2022). CYPs are also known to be the key drivers of alkaloid chemical diversification in plants [35,45]. We identified 184 CYP genes in the transcriptomes of *C. robustum.* Nelson and Schuler [46] presented CYP gene copies in 15 sequenced/representative plant genomes, including eight flowering plant species, such as lotus, *Arabidopsis*, papaya, grape, poplar, soybean, rice, and *Brachypodium*. Among those eight flowering plant species, only lotus (172 copies) and papaya (142 copies) had a lower number of CYP gene copies than our *C. robustum* result. The number of CYP copies in *C. robustum* was lower than that in the *Berberis* species, which belongs to Berberidaceae as *Caulophyllum.* The CYP copy numbers were 257 in *Berberis koreana*, 255 in *B. thunbergii*, and 253 in *B. amurensis* [36]. Nelson [47] grouped 16,000 plant CYPs into 277 families on the basis of 40% or higher sequence sharing so that we also adopted his suggestion on the identification of CYP families. In *C. robustum*, we identified 56 CYP families and the *Berberis* species had 31 families [36], in which 18 families shared between the *Caulophyllum* and *Berberis* species. The 31 CYP families in the *C. robustum* were further classified into 56 subfamilies, in which, only 16 subfamilies were shared with the *Berberis* species, implying that CYP differentiation was evident in subfamilies, and occurred even within Berberidaceae.

Among the plant CYP families, CYP80, CYP82, and CYP719 were known to be involved in berberine synthesis [34,48]. CYP80 and CYP719 were present, but CYP82 was not identified in the transcriptomes of *C. robustum* or the three *Berberis* species [36]. CYP80B3 mediates the conversion of (*S*)-*N*-methylcoclaurine to (*S*)-3′-hydroxy-*N*-methylcoclaurine, which is an intermediate BIA molecule in the synthesis of diverse morphine, sanguinarine, noscapine, and berberine [34]. The subfamily enzymes of CYP719 are involved diverse BIAs including berberine. However, CYP82 is not in the pathway for berberine synthesis. Thus, the *Caulophyllum* and *Berberis* species may not carry the CYP82 gene. Berberine is a major BIA molecule in the genera *Berberis* [16,36] and *Caulophyllum* [currnt study]. Roy et al. demonstrated that blocking the pathway leading to other BIA molecules results in the effective biosynthesis of berberine in *B. koreana* [21]. We identified one copy each of CYP80B3 and CYP719A21 in the transcriptomes of *C. robustum*, whereas *B. koreana* had five copies of CYP80 and two copies of CYP719. The berberine content in *C. robustum* leaf (99.95 ± 0.5 μg/g) is about five times higher than that in *B. koreana* leaf (19.2 ± 0.5 μg/g). Thus, the single copies of CYP80 and CYP719 may be sufficient to synthesize berberine in *C. robustum*, but this may require further study.

In the phylogenetic analysis, the 19 CYP80s consisted of CYP80B1, CYP80B2, and CYP80B3 subfamilies. The 21 CYP719s consisted of CYP719A1, CYP719A2, CYP719A6, CYP719A7, and CYP719A21 subfamilies. The CYP80 and CYP719 families were distinguished by their sequences, as shown by the formation into two separate clades. However, the subfamily assignment within the family was not unequivocal as sequences are highly similar among the members within a family. Many of these CYP enzymes were not annotated into subfamilies in the NCBI database; therefore, we carried out a BLAST analysis individually with the unidentified subfamily CYP enzymes. We then assigned the subfamily sequence sharing of 40% or higher in the BLAST analysis using the suggestion of Nelson [47]. However, we were unable to assign the *P. nudicaule* CYP719 into a subfamily, as it did not have a CYP719 subfamily sequence matching higher than 40%. CYP80 and CYP719 were specifically involved in the biosynthesis of berberine, and those species used in the phylogenetic analysis were all berberine-producing medicinal plants, except *C. micranthum,* which has been used as a medicinal plant and contains various phytochemicals, including alkaloids [49]. However, its alkaloids were not further characterized. Our result showed that most of the CYP80 and CYP719 subfamilies were differentiated within plant families, except the CYP members in the II-I and III-I groups. Moreover, the CYPs from the same species were placed in the deepest branch clade, implying that CYP gene duplication occurred within species to increase the copy numbers. Gene duplication is important for gene family differentiation, and many genes involved in the synthesis of secondary metabolites are present in gene families [21].

## 5. Summary

*Caulophyllum* species have traditionally been used in herbal medicine by native Americans. The roots and rhizomes of *C. robustum*, Asian blue cohosh, have also been widely used in folk medicine in China. Various phytochemicals have been reported as being present in this plant, but berberine has not been identified until current study. We found high berberine content in *C. robustum* leaves; therefore, we carried out a transcriptome analysis to characterize the berberine biosynthesis genes in *C. robustum*. We obtained 41,094 transcripts that were annotated and characterized by GO analysis. Vis-à-vis comparison of the number of genes expressed in leaf, stem, and fruit was conducted with GO annotation, in which, more genes were expressed in leaf compared to fruit and stem. We isolated all the genes involved in the berberine synthesis pathway in the transcriptomes of *C. robustum*, and the compared gene copies in six other berberine-producing plants. Phylogenetic analysis was then carried out with NCS, BBE, methyltransferases, CYP80 and CYP719. In the analyses of NCS and BBE enzymes, different species were mixed in subclades of their phylogenetic tree to support the monophyletic origins of both enzymes. There were four methyltransferases, 6OMT, CNMT, 4OMT, and SOMT, involved in berberine synthesis. In the phylogenetic analysis, the CNMTs and SOMTs were clustered in their own subclade, and the 4OMTs and 6OMTs split into two groups, which means that the duplication of 4OMTs and 6OMTs occurred early in the evolution of the plant. We identified 184 CYPs, in which, 132 CYPs were classified into 31 families, with the rest not classified. CYP80 and CYP719 are involved in berberine synthesis, and we identified one copy each of CYP80 and CYP719. The phylogenetic analysis revealed that the differentiation into subfamilies in both CYP80 and CYP719 occurred mostly at the plant family level. The phylogenetic tree also revealed that the copy number increment occurred at the species level.

## Figures and Tables

**Figure 1 plants-12-01483-f001:**
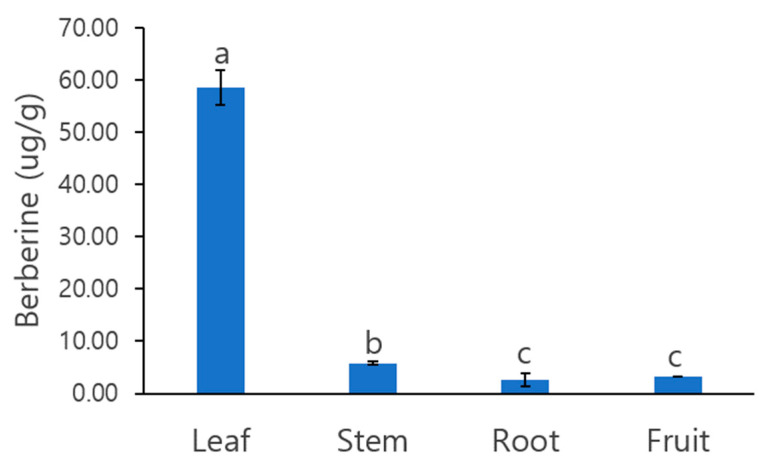
Berberine content in the four organs of *C. robustum*. Data followed by a different letter indicate significant differences between samples according to one-way ANOVA/Duncan test (*p* < 0.05).

**Figure 2 plants-12-01483-f002:**
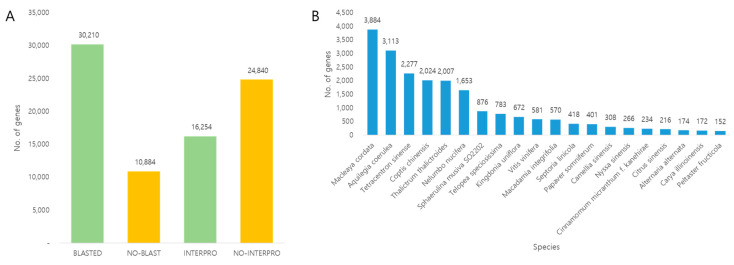
(**A**) Mapping of the transcripts to different protein databases. (**B**) List of species matching the *C. robustum* transcriptomes to other transcriptomes.

**Figure 3 plants-12-01483-f003:**
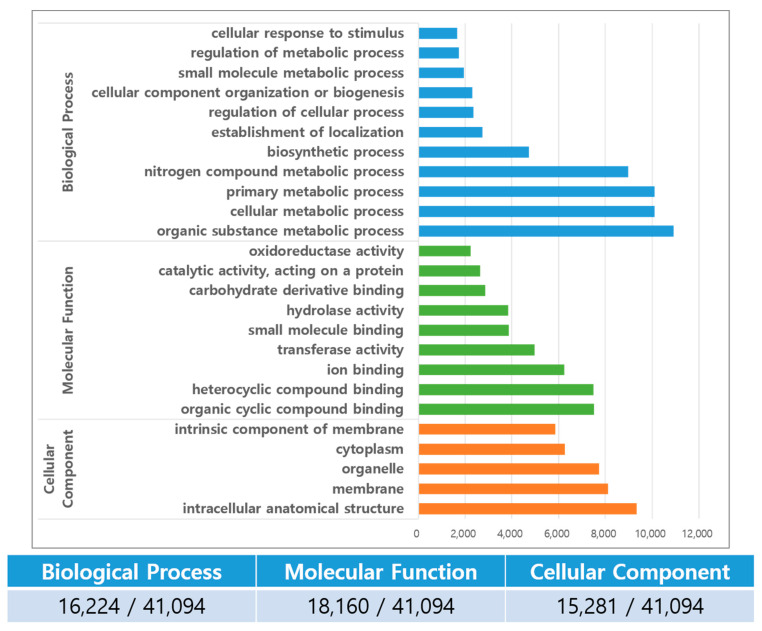
GO analysis of the *C. robustum* transcriptomes. The numbers at the bottom represent the number of transcripts in the three GO categories from whole set of transcriptomes.

**Figure 4 plants-12-01483-f004:**
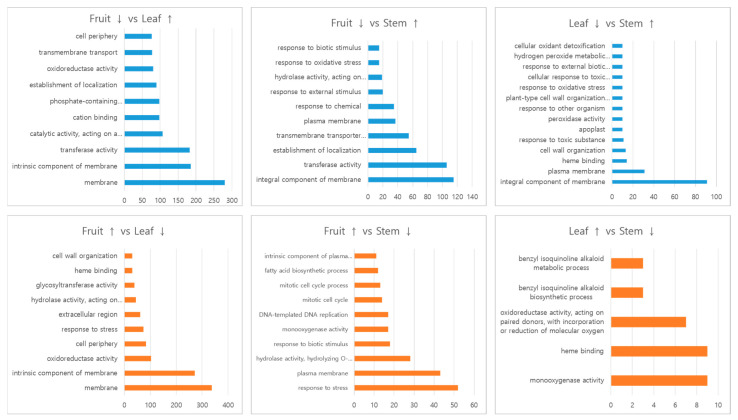
GO analysis of the differently expressed genes between tissues. The numbers in the X-axis represent the number of transcripts.

**Figure 5 plants-12-01483-f005:**
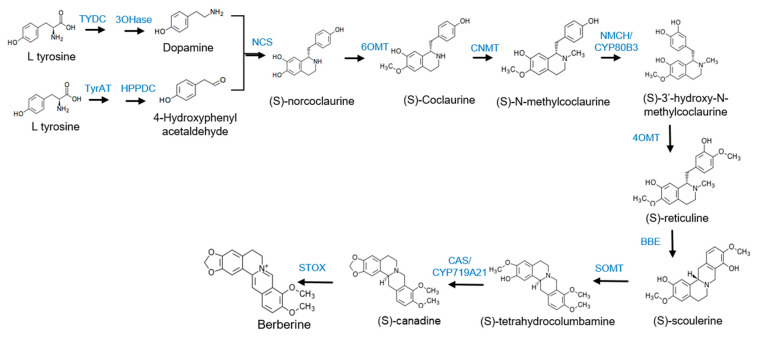
Berberine synthesis pathway. TYDC: tyrosine decarboxylase; 3OHase: tyrosine/tyramine 3-hydroxylase; TyrAT: tyrosine aminotransferase; HPPDC: 4-hydroxyphenylpyruvate decarxylase; NCS: (*S*)-norcoclaurine synthase; 6OMT: (*RS*)-norcoclaurine 6-*O*-methyltransferase; CNMT: (*S*)-coclaurine N-methyltransferase; NMCH/CYP803: *N*-methylcoclaurine 3′-monooxygenase; 4OMT: 3′hydroxy-N-methyl-(*S*)-coclaurine 4′-*O*-methyltransferase; BBE: berberine bridge enzyme; SOMT: (*S*)-scoulerine 9-*O*-methyltransferase; CAS/CYP719A21: canadine synthase; STOX: (*S*)-tetrahydroprotoberberine oxidase.

**Figure 6 plants-12-01483-f006:**
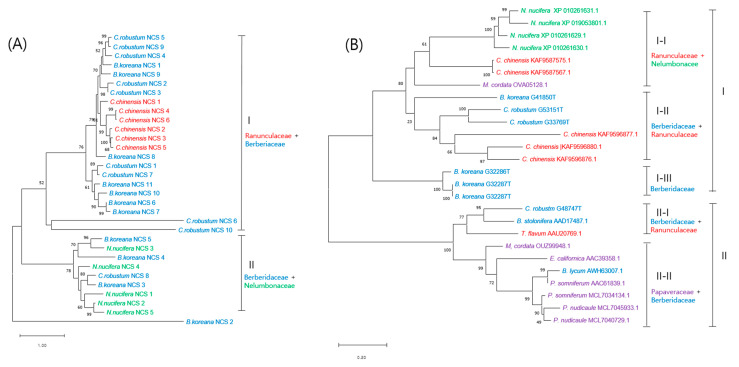
Phylogenetic trees of: (S)-norcoclaurine synthase (NCS) in (**A**); and berberine bridge enzyme (BBE) in (**B**). Species—*B. lyceum*: *Berberis lyceum*; *B. koreana*: *Berberis koreana*; *C. chinensis*: *Coptis chinensis*; *C. robustum*: *Caulophyllum robustum*; *E. californica*: *Eschscholzia californica*; *M. cordata*: *Macleaya cordata*; *N. nucifera*: *Nelumbo nucifera*; *T. flavum*: *Thalictrum flavum*; *P. somniferum: Papaver somniferum*; *P. nudicaule*: *Papaver nudicaule*. The scale bar represents the phylogenetic distance derived from substitution per site in the sequence.

**Figure 7 plants-12-01483-f007:**
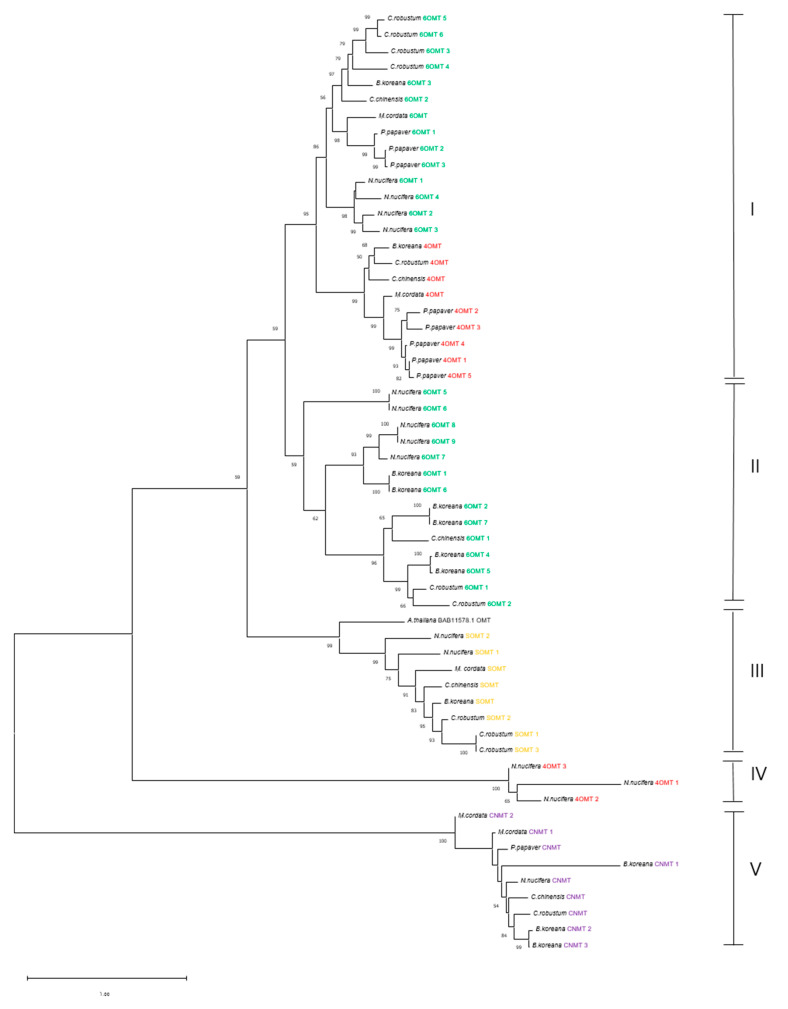
Phylogenetic tree of methyltransferases. Species—*B. lyceum*: *Berberis lyceum*; *B. koreana: Berberis koreana*; *C. chinensis*: *Coptis chinensis*; *C. robustum*: *Caulophyllum robustum*; *E. californica*: *Eschscholzia californica*; *M. cordata*: *Macleaya cordata*; *N. nucifera*: *Nelumbo nucifera*; *T. flavum*: *Thalictrum flavum*; *P. somniferum*: *Papaver somniferum*; *P. nudicaule*: *Papaver nudicaule*. The scale bar represents the phylogenetic distance derived from substitution per site in the sequence.

**Figure 8 plants-12-01483-f008:**
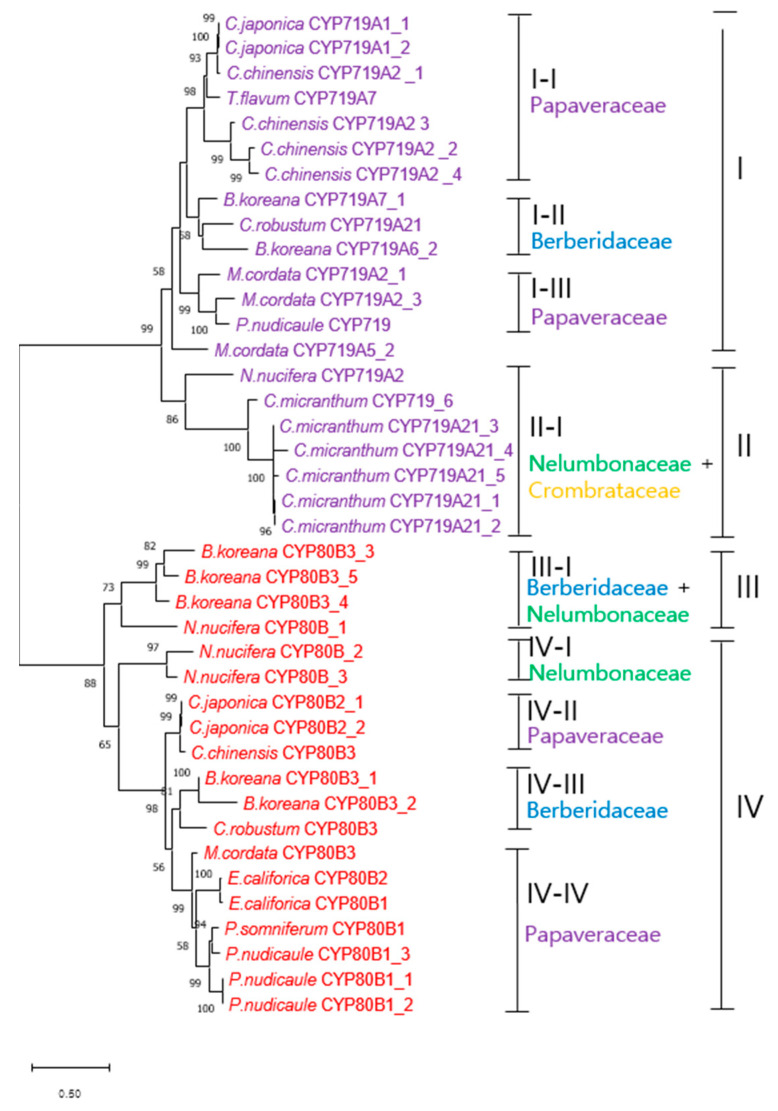
Phylogenetic tree of CYP80s and CYP719 from various plants. Species—*B. lyceum*: *Berberis lyceum*; *B. koreana*: *Berberis koreana*; *C. chinensis*: *Coptis chinensis*; *C. japonica*: *Coptis japonica*; *C. robustum*: *Caulophyllum robustum*; *E. californica*: *Eschscholzia californica*; *M. cordata*: *Macleaya cordata*; *N. nucifera*: *Nelumbo nucifera*; *T. flavum*: *Thalictrum flavum*; *P. somniferum*: *Papaver somniferum*; *P. nudicaule*: *Papaver nudicaule*; *C. micranthum*: *Combretum micranthum*. The scale bar represents the phylogenetic distance derived from substitution per site in the sequence.

**Table 1 plants-12-01483-t001:** Plant species used for mining BIAs biosynthetic genes.

Species	Order	Family	Common Name	Analyses *
*Caulophyllum robustum*	Ranunculales	Berberidaceae	Asian blue cohosh	1, 2, 3, 4, 5
*Berberis koreana*	Ranunculales	Berberidaceae	Barberry	1, 2, 3, 4, 5
*Berberis lycium*	Ranunculales	Berberidaceae	Barberry	3, 4
*Berberis stoliniferum*	Ranunculales	Berberidaceae	Barberry	3
*Combretum micranthum*	Myrtales	Combretaceae	Kinkeliba	5
*Coptis chinensis*	Ranunculales	Ranunculaceae	Chinese goldthread	1, 2, 3, 4, 5
*Coptis japonica*	Ranunculales	Ranunculaceae	Chinese goldthread	5
*Thalictrum flavum*	Ranunculales	Ranunculaceae	Common meadow-rue	3, 4
*Papaver nudicaule*	Ranunculales	Papaveraceae	Iceland poppy	1, 3, 4, 5
*Papaver somniferum*	Ranunculales	Papaveraceae	Opium poppy	3, 4, 5
*Eschscholzia california*	Ranunculales	Papaveraceae	California poppy	3, 4, 5
*Macleaya cordata*	Ranunculales	Papaveraceae	Plume puppy	1, 3, 4, 5
*Nelumbo nucifera*	Proteales	Nelumbonaceae	Sacred lotus	1, 2, 3, 4, 5
*Cinnamon micranthum*	Laurales	Lauraceae	Cinnamon	5

* 1: Mining whole genes in the berberine synthesis pathway. 2: (*S*)-norcoclauline synthase (NCS) phylogenetic analysis. 3: Berberine bridge enzyme (BBE) phylogenetic analysis. 4: Methyltransferase phylogenetic analysis. 5: CYP phylogenetic analysis.

**Table 2 plants-12-01483-t002:** Number of up- or down-expressed genes between tissues in vis-à-vis comparison.

	Fruit vs. Leaf	Leaf vs. Stem	Fruit vs. Stem
Up	761	326	536
Down	1135	105	768

**Table 3 plants-12-01483-t003:** Copies of the berberine synthesis genes in *C. robustum*, *B. koreana*, *N. nucifera*, *P. nudicaule*, *C. chinensis*, and *M. cordata*.

Enzymes	EC Numbers	*C. robustum*	*B. koreana*	*N. nucifera*	*P. nudicaule*	*C. chinensis*	*M. cordata*
TYDC	4.1.1.25	1	2	5	10	5	12
3OHase	1.14.16.2	1	2	1	4	3	*
TyrAT	2.6.1.5	2	1	8	3	3	4
HPPDC	4.1.1.80	1	2	5	10	5	12
NCS	4.2.1.78	10	11	5	-	6	1
6OMT	2.1.1.128	8	7	9	3	3	1
CNMT	2.1.140	1	3	1	1	1	2
NMCH/CYP80B3	1.14.13.71	1	5	4	3	1	1
4OMT	2.1.1.16	1	1	4	5	1	1
BBE	1.21.3.3	4	5	4	3	5	5
SOMT	2.1.1.247	3	1	2	1	1	1
CAS/CYP719A21	1.14.21.5	1	2	1	2	4	3
STOX	1.3.3.8	2	1	3	-	1	6

TYDC: tyrosine decarboxylase; 3OHase: tyrosine/tyramine 3-hydroxylase; TyrAT: tyrosine aminotransferase; HPPDC: 4-hydroxyphenylpyruvate decarboxylase; NCS: (*S*)-norcoclaurine synthase; 6OMT: (*RS*)-norcoclaurine 6-*O*-methyltransferase; CNMT: (*S*)-coclaurine N-methyltransferase; NMCH/CYP803: *N*-methylcoclaurine 3′-monooxygenase; 4OMT: 3′hydroxy-N-mthyl-(*S*)-coclaurine 4′-*O*-methyltransferase; BBE: berberine bridge enzyme; SOMT: (*S*)-scoulerine 9-*O*-methyltransferase; CAS/CYP719A21: canadine synthase; STOX: (*S*)-tetrahydroprotoberberine oxidase.

**Table 4 plants-12-01483-t004:** CYP monooxygenase families in *C. robustum*.

Family	Number of Subfamilies	Genes (Number of Genes)	Number of Genes
CYP3	1	CYP3a3 (1)	1
CYP6	1	CYP6a2 (1)	1
CYP52	1	CYP52A13 (1)	1
CYP71	11	CYP71A1 (13), CYPA2 (2), CYP71A9 (1), CYPAU50 (4), CYP71A19 (1), VYP71B34 (1), CYPBE30 (1), CYP71D8 (1), CYPD9 (1), CYP71D11 (1), CYP71B10 (1)	27
CYP72	2	CYP72A (8), CYP72A219 (14)	22
CYP76	1	CYP76A1 (2)	3
CYP77	2	CYP77A2 (1), CYP77A3 (1)	2
CYP78	3	CYP78A5 (2), CYP78A7 (6), CYP78A9 (2)	10
CYP79	2	CYP79A68 (1), CYP89A2 (1)	2
CYP80	1	CYP80B3 (1)	1
CYP81	2	CYP81Q32 (3), CYP89Q39 (1)	4
CYP82	2	CYP82A3 (1), CYP82C4 (2), CYP82D47 (1)	2
CYP84	1	CYP84A1 (3)	3
CYP85	1	CYP85A (3)	3
CYP86	3	CYP86A1 (1), CYP86A8 (1), CYP86B1 (2)	4
CYP87	1	CYP87A3 (2)	2
CYP89	1	CYP89A2 (3)	3
CYP90	2	CYP90A1, CYP90B1	2
CYP94	5	CYP94A1 (1), CYP94A2 (6), CYP94B3 (2), CYP94C1 (2), CYP94C1 (1)	12
CYP97	1	CYP97B2	1
CYP98	1	CYP98A2 (2)	2
CYP313	1	CYP313a2	1
CYP704	2	CYP704B1 (1), CYP704C1 (3)	4
CYP709	1	CYP709B2 (1)	1
CYP711	1	CYP711A1 (4)	4
CYP714	1	CYP714 (3)	3
CYP716	1	CYP716B1 (4)	4
CYP719	1	CYP719A1 (1)	1
CYP724	1	CYP724B1 (3)	3
CYP734	1	CYP734A1 (2)	2
CYP736	1	CYP736A2 (1)	1
Not classified			52
Total	56		184

Note: The genes highlighted in red are the ones shared between *C. robustum* and *B. koreana*.

## Data Availability

The raw data of the current research are linked in the http://nbitglobal.com/caulophyllum/ (accessed on 25 January 2023).

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
