# Peer review of "Isolation and Characterization of the Genes Involved in the Berberine Synthesis Pathway in Asian Blue Cohosh, Caulophyllum robustum"

_plants, 2023, doi:10.3390/plants12071483_

Round 1
Reviewer 1 Report
Comments for the paper entitled “Isolation and Characterization of the Genes Involved in the Berberine Synthesis Pathway in Asian Blue Cohosh, Caulophyllum robustum”
The paper deals with fundamental research of the berberine synthesis pathway in Asian Blue Cohosh, Caulophyllum robustum. The authors have attempted to see the transcriptomic analysis to elucidate the level of genes/enzymes involved in the berberine synthesis pathways. The authors also confirm the berberine content level in the leaves, stems and fruits of C. robustum. They tried correlating their transcriptomic data with the other closer or same family species. They also elucidated the key enzymes involved during the berberine synthesis and showed their phylogenetic analysis with their closer species and family. As per the authors, this is the first report of a berberine study in C. robustum. The idea of paper seems nice. The paper presentation is also good, and I have no objection to publishing this research in its current form after some significant corrections.
Nevertheless, the manuscript is composed well but needs some major revision before its final publication in this journal. Specific comments and possible suggestions section-wise are appended below
Abstract
Concise and well define
Introduction
Concise and well define
Material methods and results
Ok
Discussion
Needs major revision, the discussion is looking very lengthy and boring authors need to sort the discussion parts; in many places, authors should avoid the repeated introduction and results in the discussion, for example
· Line 396 to 400 is the repeated sentence of the introduction
· Lines 409 to 412 suppose to be part of the results.
· Discussion of BIA, NCS, CNMT and OMT is very lengthy, it needs to be shorter.
· Discussion of CYP also seems very lengthy and tedious; authors should be concise in a single para and avoid representing the repeating results part along with discussion like it appeared that phylogenetic study results have been reproduced repeatedly.
· In precise, the philosophy of the discussion is not looking good; authors should have briefed the discussion to avoid the repetition of introduction and results. There is a big lack of connection between the para in the discussion and it appears that every enzyme is discussed separately and there is no link between the enzymes. Because of this region discussion has not appeared in uniform and in flow. However, the work concept is nice and could be discussed nicely.
Summary
· Summary should be concise and should not include results of the findings, for example, lines 587-592.
Minor comments
· Figure 4 is not reproducible; provide another figure; it appears very blurry
· Figure 8 should be the supplementary figure
· Line 372 amd to and
· Line 373 res ??
· Line 376 CYP719) ??
· line 388 plants. Species
· Species names should always be in italics see lines 388 to 392
Author Response
Overall: Thank you for your favoring comments.
- Line 396 – 400: repeated sentences of the introduction: corrected (see the first paragraph of Discussion)
- Line 409 – 412: We changed the sentence as “We found that the leaves contained considerably more berberine than other tissues.”
- Lengthy discussion of the BIA, NCS, CNMT, and OMT: we deleted some redundant sentences.
- Summary: we revised accordingly.
- Figure 4: revised.
- Figure 8: we assigned it to Supplementary Figure 1.
- Line 373: “amd”: corrected as “and”
- Line 373: “res and”: revised as “rest and”
- Line 376 CYP): we corrected as 376 CYP
- Line 388-392: we checked them all italicized.
Reviewer 2 Report
Dear authors, I appreciate the amount of work done. The work is interesting and the transcriptome data is fine for publication. However, I have some comments (2 minors and 3 majors):
Please put vertical axis names in Figures 2A and 2B.
In table 3, please mention, what does the “*” signify?
The transcriptome data must be validated by real-time PCR. Please check the expression pattern of the related genes.
Also, I recommend conducting biochemical assays (if possible) for CYP450 monooxygenases.
The summary section (point 5) must be direct to the point. There are many points (e.g. number of transcripts etc.) which have already been discussed and written in the previous sections.
Author Response
- Figure 2 modification: we have done accordingly.
- Table 3: we corrected the “*” in the table.
- Validation of the expression by qPCR. Thank you for your suggestion. I agree that validation by qPCR can add solid expression results. However, it is often challenged for robustness of the validation between RNA-seq and qPCR (or microarray) [“Do results obtained with RNA-sequencing require independent verification?” Biofilm 43 (2021) 100043. DOI 10.1016/j.bioflm.2021.100043]. In our analysis, we are giving differences of the number of genes expressed different tissues in conjunction with GO analysis (Table 2 and Figure 4). We did not perform the differential expression of individual genes in different tissues. Thus, we changed the subtitle to “Differences of the genes expressed among leaves, stem, and fruit in C. robustum” to avoid the confusion. We hope this does not degrade much of study.
- Biochemical assay for CYP450. This is a good suggestion. However, we focus on the phylogenetic differentiation of the CYPs rather than doing for biochemical assay of the CYPs which might be a subject for another study.
5. Summary section: We revised the summary.
Reviewer 3 Report
The authors of the manuscript titled “Isolation and Characterization of the Genes Involved in the Berberine Synthesis Pathway in Asian Blue Cohosh, Caulophyllum robustum.” The current report contains the characterization of berberine synthesis genes in C. robustum along with a comparative analysis of those genes with genes in other berberine-producing plants.
General comments
Overall, the study is well-designed and presented in a good way.
Abstract
The authors elaborated the abstract in a good way. However, some points need to be addressed to improve this section.
Kindly, add the summary of the current study in the last paragraph of the abstract.
Introduction
This section is also well written but the authors are requested to study the manuscript carefully and italicize genes and scientific names.
Materials and Methods
The authors are requested to add a reference for the “Berberine extraction and quantification” section
Results and Discussion
These sections are well written.
Summary
The summary section of the manuscript should include a summary of the results and discussion, followed by the author’s perspective on how this study will contribute to the current scientific knowledge. Furthermore, the authors could elaborate on the expected amplification of their work moving forward and explain the relevance of conducting this research in the first place. Providing such insights would enhance the impact and overall value of the manuscript. Therefore, the authors are requested to add future perspective of the current study.

Author Response
Thank you for your favoring comment. We appreciate your time to review our manuscript.
Round 2
Reviewer 1 Report
accepted in the current revised form
Author Response
Reviewer 1
Quality of English:
Answer: The manuscript was edited by MDPI English editing services. After edited, title of the manuscript was changed the current title “Isolation and characterization of the genes involved in the berberine synthesis pathway in Asian blue cohosh, Caulophyllum robustum”. Please see the certificate from the MDPI AUTHORservices.
Thank you.
Regards,
On behalf of the co-authors
Nam-Soo Kim, Ph.D

Reviewer 2 Report
Major comment:
The authors have modified the summary section, however, I do not agree with the fact that “RNA-seq results need not be validated by real-time PCR”. There are no doubts about the robustness of the RNA-seq transcriptome data, but qRT-PCR is still considered the gold standard for gene expression analysis. NGS provides a great tool for analyzing a large number of genes to narrow down the research interest, but I suggest doing qRT-PCR validation studies.
Minor comments:
1. Please correct the vertical axis name in Figure 1.
2. All the gene names and scientific names of the species must be in italics, throughout the manuscript.
Author Response
- Quality of English:
Answer: The manuscript was edited by MDPI English editing services. After edited, title of the manuscript was changed the current title “Isolation and characterization of the genes involved in the berberine synthesis pathway in Asian blue cohosh, Caulophyllum robustum”. Please see the certificate from the MDPI AUTHORservices.
- Validation by qPCR
Answer: I agree with your opinion that the qRT-PCR is still considered as the gold standard expression analysis. The read count differences in RNA-seq results will be validated by the qRT-PCR analysis on the expression level differences of specific genes in different tissues. As I have explained in our previous response, however, we are giving the differences of the number of genes expressed in different tissues with GO analysis. Table 2 describes the number of up- or down-expressed genes between tissues in vis-à-vis comparison and figure 2 shows their functional profiles by GO analysis. We did not perform the differential expression of individual genes between different tissues, which should be followed by qRT-PCR analysis for validation, as you suggested.
- Please correct the vertical axis name in Figure 1.
Answer: Thank you for your point. I googled the definition of ppm, which says “PPM or Parts Per Million is a unit used to describe very small concentrations of a substance in a larger solution.” Thus, ppm is inappropriate to describe in our case. We changed it as “Berberine (ug/g)”.
- All the gene names and scientific names of the species must be in italics, throughout the manuscript.
Answer: We italicized all the gene names and species scientific names including Figures 6, 7, and 8.
Thank you.
Regards,
On behalf of the co-authors
Nam-Soo Kim, Ph.D

Reviewer 3 Report
I accept the manuscript in the current form after revision.
Author Response
Thank you!
Round 3
Reviewer 2 Report
Dear authors, many thanks for the explanations. I am satisfied with the explanations. The manuscript can be accepted.